# OAR: Training Quantization-Friendly Object Detectors via Outlier-Aware Restriction

## Abstract

Model quantization is widely employed to reduce computational resource usage during inference, often in conjunction with specialized hardware system for acceleration. While modern object detectors perform well at compact bit-widths (e.g., 8-bit), further quantization to ultra-low bit-widths (e.g., 4 or 3 bits) remains challenging. We identify the presence of outliers in the statistical distribution of activations in pre-trained detectors as a key obstacle, as such outliers expand the dynamic range and increase quantization error. Moreover, we observe significant numerical discrepancies in activation outliers across the task branches of the detection head, potentially leading to imbalanced sub-task performance after quantization. To address these issues, we propose Resonant Shrinkage loss and Output Adaptation to suppress activation outliers during pre-training. Additionally, we introduce Edge-Aware KURE loss to enhance the robustness of detector weights under quantization. All components are applied during the pre-training phase, producing detectors that are more quantization-friendly without altering training hyperparameters, while also reducing quantization sensitivity disparities between task branches. Our training framework is compatible with existing state-of-the-art quantization methods and delivers improved performance. Notably, even with naive post-training quantization requiring only a small calibration set, our 4-bit quantized ATSS model achieves 35.39% mAP, outperforming the original quantized version by 3.03%. Code will be released soon.

## 1 Introduction

Deep learning models have emerged as the main solution for object detection tasks, achieving impressive results in the fields of autonomous driving and medical image analysis. Given that many applications running these models operate on edge devices (e.g., phones, In-Vehicle Infotainment, and drones), striking a balance between the accuracy, latency, energy efficiency, and size of these computationally intensive models is imperative for their deployment. Model quantization is an effective technique for achieving a trade-off between these conflicting requirements. It involves mapping each floating-point number to a set of discrete quantization levels based on the bit-width requirement. Then the quantized detectors can be executed on specialized platforms that support low-bit computation. When the quantization process is extended to lower bit precision levels, such as 4-bit or 2-bit, there is a notable degradation in the performance of the model. The bit resolution of quantization is inversely proportional to the range of the statistical data (i.e., weights and activations) of the model, which consequently affects the performance of the quantized model. Since outliers tend to increase the range of the statistical data, they are harmful to quantization-friendly models.

During quantization procedure for object detectors, we observe more pronounced differences in activation ranges between layers at the same depth across different task branches in the detection head. Figure 1(a) demonstrates this problem by displaying the statistical results of the activations and weights for each layer in the classification and regression branches at the head structure. The differences in activations at the later depths are highly significant, although the weight differences are minimal. As the statistical edge and statistical quartile of a data tensor increase, the intervals between the quantization levels after quantization become larger, thereby producing more quantization error. Consequently, the accumulated quantization error of the corresponding task branch also increases, resulting in graver quantization sensitivity to the sub-task and subsequently degrading

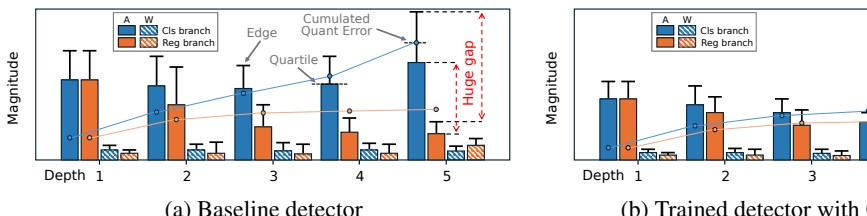

(a) Baseline detector         (b) Trained detector with OAR (Ours)

Figure 1: Illustration of the statistics of activations at each same depth in the head structure. "A" and "W" represent activations and weights, respectively. In (a), we only present the convolutional layers for simplicity.

the sub-task performance. This problem commonly arises in modern detectors that adopt a head structure consisting of independent and parallel branches (i.e., task-specific sub-nets).

Therefore, mitigating activation variability in the detection head is both necessary and critical. We propose addressing this by constraining outliers during training from scratch. Specifically, we introduce the **Resonant Shrinkage loss**, which regularizes activation boundaries while maintaining batch-wise consistency to reduce quantization sensitivity. Additionally, we hypothesize that applying Softmax or Sigmoid to the final outputs may amplify activation ranges across branches. To address this, we introduce **Output Adaptation**, which applies a learnable scaling factor to each branch output to constrain the output distribution range. Finally, inspired by KURE loss Shkolnik et al. (2020), we propose the **Edge-Aware KURE loss** to refine the statistical distribution of weights. Hereafter, we refer to our proposed training framework as Outlier-Aware Restriction (OAR). OAR can be effortlessly incorporated into the training pipeline of any object detector, serving to improve the final performance of existing quantization techniques. As shown in Figure 1(b), detectors trained with OAR exhibit more consistent activation statistics within the head, leading to improved robustness across different bit-widths and quantization strategies under identical training settings. To the best of our knowledge, this is the first work to investigate the sensitivity discrepancies among task branches in detector heads, which are notably reflected in the statistical inconsistencies of activations at the same depth.

In summary, our contributions are as follows:

- We introduce the **Resonant Shrinkage loss**, which constrains activation boundaries during training while maintaining consistency of activations across batches, thereby reducing sensitivity to quantization.

- We propose **Output Adaptation**, which applies a learnable scaling factor to the output of each task branch, effectively normalizing the output distribution range.

- To further enhance quantization robustness, we present the **Edge-Aware KURE loss**, designed to refine the statistical distribution of weights.

- Extensive experiments on the MS COCO Lin et al. (2014) dataset across various detector architectures validate the effectiveness of our proposed method.

## 2 RELATED WORKS

### 2.1 QUANTIZATION-AWARE TRAINING

Quantization-aware training (QAT) adjusts the quantization noise during the training phase using fully labeled datasets. Straight-Through Estimator (STE) Bengio et al. (2013) addresses the issue of zero gradients generated by the rounding function during backpropagation via a gradient estimation technique. While some works Choi et al. (2018); Esser et al. (2020); Bhalgat et al. (2020); Jain et al. (2020); Nagel et al. (2022) explore different quantization strategies based on the STE, others Gong et al. (2019); Lee et al. (2021); Shin et al. (2022) refine the backpropagation of gradients during simulated quantization, particularly focusing on improving the gradient estimation for rounding operations rather than directly relying on the STE.

## 2.2 POST-TRAINING QUANTIZATION

Post-training quantization (PTQ) frequently ascertains the quantization scaling factors using a limited set of calibration data by performing a straightforward search within the parameter space Banner et al. (2019); Choukroun et al. (2019); Wu et al. (2020); Zhao et al. (2019). The metrics employed in the search for optimal scaling factors encompass the Mean Squared Error (MSE) distance Choukroun et al. (2019) and the cosine distance Wu et al. (2020). In recent years, several methods have been proposed to reconstruct rounding values Nagel et al. (2020); Li et al. (2021a); Wei et al. (2022); Liu et al. (2023). These reconstruction-style algorithms involve the utilization of gradients but still only need a minimal amount of unlabeled data.

## 2.3 REGULARIZATION FOR QUANTIZATION

Regularization is one of the crucial optimization techniques for minimizing generalization errors in neural network training. Recent studies have explored the utilization of weight regularization to enhance the robustness and performance of network quantization. Shkolnik et al. (2020); Alizadeh et al. (2020); Han et al. (2021); Xu et al. (2023); Kundu et al. (2024).

The aforementioned works have discussed the adjustment of the statistical distribution of weights, but they do not address how to adjust the statistical distribution of activations to make them more suitable for quantization. In contrast, our work focuses on gaining the quantization robustness of both activations and weights, and particularly alleviates the differences in activation distributions among different branches within the detection head.

# 3 METHODOLOGY

## 3.1 QUANTIZATION PRELIMINARY

Quantizing a neural network model can be described as a finite affine transformation. Given a full-precision vector $\mathbf{X} = [x_0, x_1, ..., x_{n-1}]$ and a bit-width $b$, the quantization function $Q_b(\cdot)$ can be expressed as:

$$\hat{\mathbf{X}} = Q_b(\mathbf{X}) \in \{q_0, q_1, ..., q_{2^b-1}\}, \tag{1}$$

where $q_i \in \mathbb{R}$ represents a quantization level, and $2^b - 1$ is the total number of quantization levels. This work adopts a symmetric uniform quantization scheme to quantize both the weight tensors and activation tensors (i.e., feature maps), which means the intervals between quantization levels are the same. Generally, a simulated quantization process involves quantization and de-quantization during forward propagation:

$$\bar{x} = clip(\lfloor \frac{x}{s} \rceil, N_{min}, N_{max}), \quad \hat{x} = s \cdot \bar{x}, \tag{2}$$

where $s$ denotes the quantization step size, and $\lfloor \cdot \rceil$ serves as the rounding operator to map floating-point values to the nearest integers. For signed data, $N_{min} = -2^{b-1}$ and $N_{max} = 2^{b-1} - 1$, while for unsigned data, $N_{min} = 0$ and $N_{max} = 2^b - 1$. Equation equation 2 essentially maps values to the integer domain through quantization, followed by de-quantization to restore the original scale. These two steps enable the effect of quantization to be simulated while maintaining the scale range of the input tensor. While performing the QAT process, the Straight-Through Estimator (STE) Bengio et al. (2013) is commonly employed to approximate the gradient of the rounding function as 1 during backward propagation. The local gradient of $\hat{x}$ with respect to $x$ can be written as:

$$\frac{\partial \hat{x}}{\partial x} = 1, \quad N_{min} \leq \frac{x}{s} \leq N_{max}. \tag{3}$$

Quantization inevitably introduces quantization error. Treat symmetric uniform quantization as an instance; the formula for calculating the quantization error is defined as follows:

$$E^Q = ||\hat{\mathbf{X}} - \mathbf{X}||_F^2, s.t. \ \hat{\mathbf{X}} \in s \times \{-2^{b-1}, ..., 0, ..., 2^{b-1} - 1\} \tag{4}$$

where $|| \cdot ||_F$ denotes the Frobenius norm. When calibrating the quantization step size $s$ using calibration data, minimizing the quantization error $E^Q$ requires the boundaries $N_{min}$ and $N_{max}$ to be relatively aligned with outliers. In other words, outliers and anomalous distributions can inflate

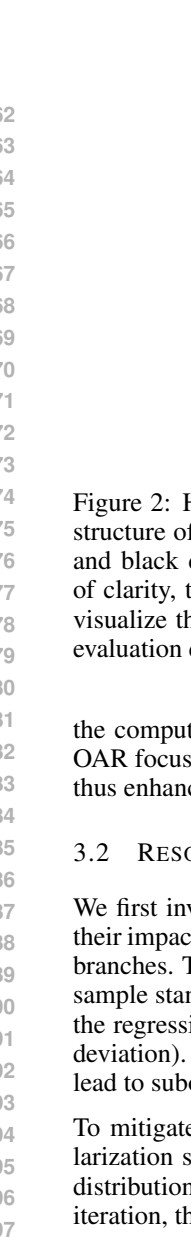
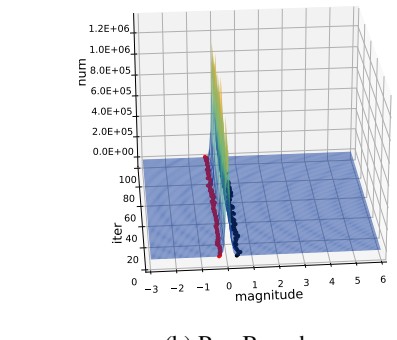

(a) Cls Branch            (b) Reg Branch

Figure 2: Histograms of inputs at the 3-th depth convolution layer, 1-th pyramid level in the head structure of a ResNet-18 He et al. (2016) backboned RetinaNet Lin et al. (2017) detector. The red and black dots represent the 0.1% and 99.9% percentiles of the data, respectively. For the sake of clarity, the tensors are not been processed by any activation functions (such as ReLU), and we visualize the inference results of only the first 100 samples from the MS COCO Lin et al. (2014) evaluation dataset.

the computation of $E^Q$, thereby adversely impacting quantization performance. This is why our OAR focuses on constructing a detector with effective outlier dampening during the training phase, thus enhancing the quantization robustness of the model from its root cause.

## 3.2 RESONANT SHRINKAGE LOSS

We first investigate the activation differences in different branches of the same depth and analyze their impact on model quantization. Figure 2 visualizes the histogram of activations for different task branches. The classification branch has a broad mountain-like overall distribution (indicating larger sample standard deviation) and more extreme outliers noticeably distant from the main body, while the regression branch exhibits a blade-like overall distribution (indicating smaller sample standard deviation). These characteristics make the classification branch more sensitive to quantization and lead to suboptimal quantization parameter selection compared to the regression branch.

To mitigate the quantization sensitivity of different task branches, we intuitively employ a regularization scheme for activations that constrains outliers within each batch while harmonizing the distribution states across different batches. Considering a $l$-th layer input $\mathbf{X}$ under the $t$-th batch iteration, the sample-wise standard deviation $\sigma$ can be obtained:

$$\mu_{t,l,i} = \frac{1}{N} \sum_{j=1}^{N} x_{t,l,i,j} \tag{5}$$

$$\sigma_{t,l,i} = \sqrt{\frac{1}{N} \sum_{j=1}^{N} (x_{t,l,i,j} - \mu_{t,l,i})^2}, \tag{6}$$

where $i$, $j$, and $N$ are used to denote the sample index of a batch, the flattened spatial position of an entry of a matrix, and the corresponding number of entries, respectively. $\sigma_{t,l,i}$ indicates the input discreteness for a sample and is also positively correlated with the size of the numerical range. The standard deviations of different samples vary. We define a boundary for the standard deviation and expect the $\sigma_{t,l,i}$ of each sample to be as close to this boundary as possible. This boundary will be updated during training, using the exponential moving average as defined below:

$$\mathcal{B}_{t,l} = \alpha \mathcal{B}_{t-1,l} + \frac{1-\alpha}{M} \sum_{i=1}^{M} \sigma_{t,l,i}, \tag{7}$$

where $\alpha$ is the weighting factor, and $M$ is the batch size for the current iteration. We set the standard $\alpha$ as 0.9. Then the formula definition of resonance loss is as follows:

$$\mathcal{L}_R^{t,l,i} = \left( \frac{\sigma_{t,l,i} - \mathcal{B}_{t,l}}{\mathcal{B}_{t,l}} \right)^2 \tag{8}$$

Note that the loss has been standardized by $\mathcal{B}_{t,l}$, which offsets the amplitude differences between different layers. During the training process, the standard deviation of each sample will continuously approach the boundary $\mathcal{B}$, and the boundary will be adaptively adjusted, ultimately achieving resonance between samples in an appropriate state. Then, to penalize the value range of the input samples, we define a shrinkage loss as:

$$\mathcal{L}_S^{t,l,i} = \frac{\max\left(|\boldsymbol{X}_{t,l,i}|\right)}{3\sigma_{t,l,i}} \tag{9}$$

We indirectly take the maximum value as punishment. The shrinkage loss eliminates the amplitude differences between layers and focuses on the outliers themselves. According to Equations 8 and 9, the Resonant Shrinkage (RS) loss for regularizing each activation is formulated by

$$\mathcal{L}_{RS}^{t,l} = \frac{1}{M}\sum_{i=1}^{M}\left(\mathcal{L}_R^{t,l,i} + \mathcal{L}_S^{t,l,i}\right) \tag{10}$$

### 3.3 OUTPUT ADAPTATION

Another reason we consider that the numerical differences between the branches are significant is the variation in post-processing mechanisms across different tasks. For regression tasks, the regression output layer typically predicts localization offsets relative to predefined anchor boxes, rather than final absolute coordinates, resulting in a much smaller range of predicted values. For classification tasks, the classification output layer directly produces raw scores, known as logits, which have not been normalized. These logits are then converted into probability by the Softmax or Sigmoid function. However, the Softmax or Sigmoid function has numerical saturation regions, meaning that if we require output probabilities close to 1 or 0, the input values must approach positive or negative infinity.

To address this issue, we advocate multiplying a learnable scaling factor at each task output to absorb potential range redundancy. We define this process as Output Adaptation (OA). Define the adaptor as $\beta$, the set of task types as $\mathbb{T}$, and the task output as $\boldsymbol{O}$. The adjusted output can be defined as:

$$\boldsymbol{O}_A^d = \beta_d\boldsymbol{O}_d \ , \ \ d \in \mathbb{T} \tag{11}$$

Note that $\beta_d$ receives gradient updates, adaptively determining the optimal scaling factor for each task branch. In practical use, most of the $\beta_d$ values are learned and function as absorption factors, reducing the range of the original outputs. Further discussion of the $\beta_d$ can be found in Section **??**.

Table 1: Performance of naive PTQ on MS COCO. N/A indicates that the performance result is close to zero. The best results in each detector are emphasized in **bold**.

| Model | With OAR | FP32 | | | INT6 | | | INT5 | | | INT4 | | |
|---|---|---|---|---|---|---|---|---|---|---|---|---|---|
| | | mAP | $AP_{50}$ | $AP_{75}$ | mAP | $AP_{50}$ | $AP_{75}$ | mAP | $AP_{50}$ | $AP_{75}$ | mAP | $AP_{50}$ | $AP_{75}$ |
| RetinaNet (ResNet-18) | ✗ | 31.73 | 49.58 | 33.43 | 31.31 | 49.03 | 33.17 | 29.63 | 47.11 | 30.80 | 23.96 | 39.38 | 24.54 |
| | ✓ | **32.01** | **50.27** | **33.78** | **31.74** | **49.95** | **33.46** | **30.86** | **48.81** | **32.63** | **27.75** | **44.86** | **29.00** |
| RetinaNet (ResNet-50) | ✗ | **36.50** | 55.40 | **39.10** | 36.01 | 54.88 | **38.25** | 34.96 | 53.68 | 37.04 | 30.28 | 47.45 | 31.61 |
| | ✓ | 36.43 | **56.06** | 38.68 | **36.13** | **55.68** | 38.15 | **35.66** | **54.93** | **37.74** | **32.71** | **51.05** | **34.62** |
| ATSS (ResNet-50) | ✗ | **39.45** | 57.61 | **42.81** | 38.86 | 56.90 | 41.97 | 37.58 | 55.37 | 40.44 | 32.36 | 48.91 | 34.66 |
| | ✓ | 39.30 | **57.62** | 42.40 | **39.06** | **57.24** | **42.06** | **38.25** | **56.23** | **41.36** | **35.39** | **52.74** | **38.05** |
| YOLOX-tiny (CSPDarknet) | ✗ | 31.78 | 49.08 | 33.81 | 27.45 | 44.37 | 29.44 | 23.88 | **40.58** | 25.28 | N/A | N/A | N/A |
| | ✓ | **32.89** | **50.15** | **34.80** | **29.35** | **46.41** | **31.22** | **24.39** | 40.48 | **25.58** | N/A | N/A | N/A |

### 3.4 EDGE-AWARE KURE LOSS

To further refine our quantization-friendly training framework, we introduce a regularization scheme for weights to enhance their robustness to quantization strategies. Inspired by KURE Shkolnik et al. (2020), we propose the Edge-Aware KURE (EA-KURE) Loss, which not only adjusts the

Table 2: Performance of Qdrop on MS COCO. N/A indicates that the performance results is close to zero.

| Model | With OAR | FP32 | | | INT4 | | | INT3 | | | INT2 | | |
|---|---|---|---|---|---|---|---|---|---|---|---|---|---|
| | | mAP | $AP_{50}$ | $AP_{75}$ | mAP | $AP_{50}$ | $AP_{75}$ | mAP | $AP_{50}$ | $AP_{75}$ | mAP | $AP_{50}$ | $AP_{75}$ |
| RetinaNet (ResNet-18) | ✗ | 31.73 | 49.58 | 33.43 | 29.39 | 46.59 | 30.71 | 25.90 | 42.49 | 26.55 | 15.51 | 28.11 | 14.99 |
| | ✓ | **32.01** | **50.27** | **33.78** | **30.41** | **48.19** | **31.78** | **27.40** | **44.70** | **28.29** | **15.88** | **28.91** | **15.26** |
| RetinaNet (ResNet-50) | ✗ | **36.50** | 55.40 | **39.10** | 34.15 | 52.70 | 36.12 | 29.93 | 47.75 | 30.90 | **17.05** | **30.51** | **16.79** |
| | ✓ | 36.43 | **56.06** | 38.68 | **34.62** | **53.66** | **36.67** | **30.70** | **49.09** | **31.86** | 16.70 | 30.12 | 16.49 |
| ATSS (ResNet-50) | ✗ | **39.45** | 57.61 | **42.81** | 36.55 | 54.19 | 39.23 | 31.96 | 48.78 | 34.17 | 17.85 | 30.19 | 18.27 |
| | ✓ | 39.30 | **57.62** | 42.40 | **37.09** | **55.02** | **39.87** | **32.90** | **50.00** | **35.11** | **18.76** | **31.37** | **19.34** |
| YOLOX-tiny (CSPDarknet) | ✗ | 31.78 | 49.08 | 33.81 | 16.19 | 30.56 | 15.70 | N/A | N/A | N/A | N/A | N/A | N/A |
| | ✓ | **32.89** | **50.15** | **34.80** | **18.34** | **33.44** | **18.34** | N/A | N/A | N/A | N/A | N/A | N/A |

Table 3: The Performance of LSQ. ① represents the utilization of HQOD in QAT phase and ② represents the utilization of OAR in pre-training phase.

| Model | Method | FP32 | | | INT2 | | |
|---|---|---|---|---|---|---|---|
| | | mAP | $AP_{50}$ | $AP_{75}$ | mAP | $AP_{50}$ | $AP_{75}$ |
| RetinaNet (ResNet-18) | LSQ | 31.73 | 49.58 | 33.43 | 23.19 | 38.47 | 23.98 |
| | +① | 31.73 | 49.58 | 33.43 | 24.91 | 40.89 | 25.90 |
| | +② | 32.01 | 50.27 | 33.78 | 25.63 | 42.02 | 26.38 |
| | +①+② | **32.01** | **50.27** | **33.78** | **26.68** | **43.20** | **27.21** |
| RetinaNet (ResNet-50) | LSQ | 36.50 | 55.40 | 39.10 | 25.02 | 41.31 | 25.88 |
| | +① | **36.50** | 55.40 | **39.10** | 27.48 | 43.96 | 28.56 |
| | +② | 36.43 | 56.06 | 38.68 | 30.11 | 48.24 | 31.70 |
| | +①+② | 36.43 | **56.06** | 38.68 | **30.91** | **48.72** | **32.56** |
| ATSS (ResNet-50) | LSQ | **39.45** | 57.61 | **42.81** | 30.62 | 47.70 | 32.13 |
| | +① | 39.45 | 57.61 | 42.81 | 33.12 | 50.39 | 35.22 |
| | +② | 39.30 | 57.62 | 42.40 | 32.45 | 50.03 | 34.45 |
| | +①+② | 39.30 | **57.62** | 42.40 | **34.05** | **51.55** | **36.21** |

distribution of weights to be more uniform-like, but also focuses on the restriction of outliers. Firstly, Kurtosis is utilized as a proxy for the probability distribution:

$$\text{Kt}\left[\mathcal{X}\right] = \mathbb{E}\left[\left(\frac{\mathcal{X} - \mu}{\sigma}\right)^4\right], \tag{12}$$

where $\mu$ and $\sigma$ are the mean and standard deviation of the random variable $\mathcal{X}$. Our EA-KURE loss is defined as:

$$\mathcal{L}_{EAK}^l = \max\left(|\boldsymbol{W}_l|\right) \cdot \left(\text{Kt}\left[\boldsymbol{W}_l\right] - \mathcal{K}\right)^2, \tag{13}$$

where $\mathcal{K}$ is the goal of distribution regularization and $\boldsymbol{W}_l$ is the weight tensor of the $l$-th layer. Referred to KURE Shkolnik et al. (2020), we keep the default setting for $\mathcal{K}$ as 1.8, which corresponds to an appropriate uniform distribution. EA-KURE loss not only unifies the overall weight distribution but also places a specific emphasis on optimizing the treatment of outliers. Moreover, the outliers also draw the importance of adjusting the weights of layers, giving priority to adjusting weights with anomalous distributions.

## 3.5 OVERALL OPTIMIZATION OBJECTIVE

In summary, the proposed framework is trained in an end-to-end manner, with the overall loss comprising both the original detector loss and customized regularization losses, as follows:

$$\mathcal{L}^t\left(\boldsymbol{W}, \beta_d \mid d \in \mathbb{T}\right) = \mathcal{L}_{Det}^t + \frac{\lambda}{L}\sum_{l=1}^{L}\left(\mathcal{L}_{RS}^{t,l} + \mathcal{L}_{EAK}^l\right), \tag{14}$$

where $\boldsymbol{W}$ and $\beta_d$ are learnable parameters, $\mathcal{L}_{Det}$ is the original detection loss based on different detectors and $\lambda$ is the regularization coefficient. Note that $\mathbb{T}$ is defined according to different detectors. For example, RetineNet Lin et al. (2017) has $\mathbb{T} = \{cls, reg\}$ for two tasks, while ATSS Zhang et al. (2020) has $\mathbb{T} = \{cls, reg, centerness\}$ for three tasks. $L$ is denoted as the number of layers

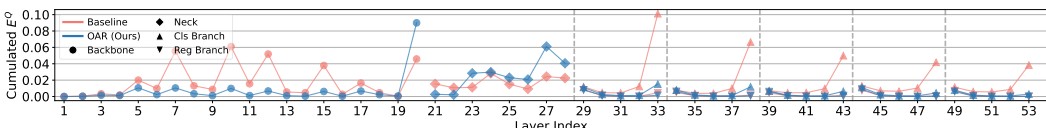

Figure 3: Visualization of the cumulated quantization error in 4-bit vanilla PTQ. Samples are from the MS COCO validation set. Indices at each depth from all pyramid levels are flattened sequentially and separated by vertical dashed lines.

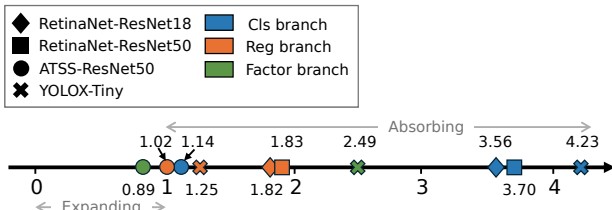

Figure 4: Analysis of adaptor $\beta$ for each task branch in different detectors.

to be quantized in the detector, indicating that our regularization scheme is not only applied to the head structure but also the backbone and neck structures to enhance the quantitation robustness of the entire model.

## 4 EXPERIMENTS

### 4.1 EXPERIMENTAL SETUP

We evaluate our method on the MS COCO 2017 Lin et al. (2014) dataset using various mainstream object detectors, including ResNet-18 He et al. (2016) backboned RetinaNet Lin et al. (2017), ResNet-50 backboned RetinaNet, ResNet-50 backboned ATSS Zhang et al. (2020), and CSPDarknet Wang et al. (2020) backboned YOLOX-Tiny Ge et al. (2021). We utilize the standard MS COCO metrics to evaluate the performance of detectors, including mean average precision (mAP), $AP_{50}$, and $AP_{75}$. The default regularization coefficient $\lambda$ is set to 1e-5 unless explicitly stated otherwise. All experiments are conducted on a single 80G A800 GPU.

**Quantization Setting.** We utilize symmetric quantization and allow Batch Normalization Folding Jacob et al. (2018). Per-channel and per-tensor quantization are used for weights and activations, respectively. The weights and inputs of all convolutional layers are quantized. The first layer of the backbone and the last layer of the head are quantized to 8 bits. We define naive PTQ as the method to solely determine quantization parameters by the calibration process. naive PTQ utilizes EMA-MSE calibration Li et al. (2021b) for activations and Min-Max calibration Li et al. (2021b) for weights. All the PTQ and QAT algorithms mentioned in experiments have been re-implemented in our codebase to enable a fair comparison under a unified quantization setting.

### 4.2 COMPARISONS TO DIFFERENT QUANTIZATION METHODS

**PTQ results.** We first evaluate the proposed method on 6-bit, 5-bit, and 4-bit naive PTQ. As shown in Table 1, models trained with OAR are more quantization-friendly than they were in their original state. On the one hand, models trained with OAR do not show significant performance degradation in fp32, and in fact, some models exhibit slight performance improvements. On the other hand, models trained with OAR achieve better performance after quantization, with the lightweight ResNet-18 backboned RetinaNet achieving nearly 4 points of mAP improvement under a 4-bit constraint. And we can gain more benefits from the lightweight YOLOX-tiny model. Table 2 presents the validation of our method on QDrop Wei et al. (2022) at lower bit levels. It can be observed that our method also achieves significant gains. Particularly, in the cases of INT4 and INT3 bits, OAR generally enhances the performance across various models. When it comes to the more difficult 2-bit data type, OAR can still improve the performance of the model in most cases.

Table 4: performance brought by different regularization algorithms from ResNet-18 backboned RetinaNet. Naive PTQ are used in 4-bit quantization.

| Method | FP32 | | | INT4 | | |
|---|---|---|---|---|---|---|
| | mAP | $AP_{50}$ | $AP_{75}$ | mAP | $AP_{50}$ | $AP_{75}$ |
| Baseline | 31.73 | 49.58 | 33.43 | 23.96 | 39.38 | 24.54 |
| +KURE | 31.36 | 49.49 | 33.27 | 26.80 | 43.42 | 28.01 |
| +R_LINF | **32.25** | **50.80** | **33.78** | 25.99 | 42.41 | 26.88 |
| +OAR | 32.01 | 50.27 | 33.78 | **27.75** | **44.86** | **29.00** |

Table 5: Performance of 2-bit naive PTQ only on the activation of layers within independent head branches of ResNet-18 backboned RetinaNet. Note that weights remain unquantized as well as other structures are not quantized. Perf Gap is defined as the mAP difference between branches after quantization. The lower the Perf Gap, the better the result.

| With OAR | Metrics | Quant Cls | Quant Reg | Perf Gap |
|---|---|---|---|---|
| ✗ | mAP | 29.53 | 30.83 | 1.30 |
| | $AP_{50}$ | 46.83 | 49.23 | 2.40 |
| | $AP_{75}$ | 30.56 | 32.84 | 2.28 |
| ✓ | mAP | 31.46 | 31.56 | **0.10** |
| | $AP_{50}$ | 49.55 | 50.20 | **0.65** |
| | $AP_{75}$ | 32.99 | 33.38 | **0.39** |

**QAT Results.** We directly validate the performance results based on LSQ Esser et al. (2020) under extreme 2-bit conditions, as shown in Table 3. We also compare our (marked as ②) results with the HQOD Huang et al. (2024) (marked as ①) algorithm, which focuses on mitigating the balance issue between classification and localization tasks in QAT quantization. Our method can significantly enhance the performance of LSQ. Moreover, When combined with HQOD, our method maximizes performance gains, yielding better results than either approach alone. This also indicates that our method has great potential for coupling and tuning with other improved algorithms.

**Comparisons to other Quantization-Aware Regularization Schemes.** Table 4 presents the 4-bit quantization results of different pretraining regularization algorithms, including OAR, KURE Shkolnik et al. (2020), and R_LINF Kundu et al. (2024). The latter two are weight-only regularization schemes, both only promoting quantization-friendly distributions for weights. The KURE scheme results in a decrease in the performance of the floating-point model, although it provides significant benefits after quantization. In contrast, the R_LINF scheme greatly enhances the floating-point performance but yields only a modest improvement after quantization. Our OAR scheme not only provides a noticeable enhancement in the mAP of the floating-point model but also delivers superior results for the quantized one.

### 4.3 Analysis on Task Branches

Table 5 compares the performance results of the baseline model with those of our method after quantizing each task branch activations individually. In the baseline model, quantizing the activations on the classification branch alone results in more significant performance degradation than the regression branch. However, the model trained with OAR not only improves the quantization sensitivity of activations in individual branches, reducing performance loss, but also narrows the quantization-induced mAP gap between the two branches by nearly 13 times, minimizing the quantization noise difference between branches.

To further support our findings, Figure 3 presents the accumulated quantization error at each layer output, including both activation and weight errors up to the current layer. Overall, OAR consistently yields lower quantization error than the baseline. However, OAR does not uniformly improve statistical metrics across all layers; in particular, its performance in the neck structure is relatively suboptimal. We attribute this to the inherent characteristics of the FPN neck, which fuses features across multiple levels via multi-input, multi-output processing, potentially compromising statistical stability due to improvements made in the backbone and head. This issue warrants further inves-

Table 6: Ablation study of the proposed OAR framework with ResNet-18 backboned RetinaNet on the MS COCO dataset. Naive PTQ are used in 4-bit quantization.

| Component | | | FP32 | | | INT4 | | |
|---|---|---|---|---|---|---|---|---|
| RS | OA | EA-KURE | mAP | $AP_{50}$ | $AP_{75}$ | mAP | $AP_{50}$ | $AP_{75}$ |
| ✓ | ✓ | ✓ | 32.01 | 50.27 | 33.78 | **27.75** | **44.86** | **29.00** |
| - | ✓ | ✓ | **32.03** | **50.45** | **33.86** | 26.18 | 42.75 | 27.08 |
| - | - | ✓ | 31.49 | 49.59 | 33.44 | 25.92 | 42.13 | 26.80 |
| - | - | - | 31.73 | 49.58 | 33.43 | 23.96 | 39.38 | 24.54 |

tigation. Nonetheless, OAR achieves notably low accumulated error at the final layer (Layer 53), effectively mitigating the impact of quantization noise on the final predictions.

To further investigate the role of Output Adaptation (OA), we visualize the scaling factors $\beta$ for each task branch, as shown in Figure 4. Most of the adaptor values exceed 1, indicating redundancy in the original output distribution ranges. Notably, all adaptor values in the classification branch are consistently greater than 1, suggesting that OA effectively absorbs the excessive activation range specific to the classification task. In the case of ATSS, the adaptor exhibits an additional behavior by expanding the range of the centerness output, indicating its flexibility in adapting to the task-specific distribution characteristics. These observations highlight the necessity of OA in dynamically adjusting the output distributions to a more quantization-friendly range, suppressing outliers, and improving statistical compactness to some extent.

### 4.4 ABLATION STUDY

We progressively remove each component and observe the corresponding performance changes, as presented in Table 6. After removing the RS loss, the performance of the quantized model significantly declines. This highlights the importance of constraining activation outliers. Further removal of OA does not lead to a significant decrease in the quantized mAP but results in a notable decline in floating-point performance. This indicates that OA correctly absorbs the distribution scale, further addressing the outlier issue. Finally, removing the EA-KURE loss also results in a significant decline in quantized performance, although the mAP of the floating-point model increases. This underscores the ability of EA-KURE to enhance the quantization robustness of weights, though relying solely on EA-KURE may potentially degrade the performance of the floating-point model.

### 5 CONCLUSION

In this work, we introduce the Outlier-Aware Restriction (OAR) training framework, which aims to enhance the quantization robustness of object detectors across various bit-widths by addressing outliers in activation and weight statistics, while minimizing the decline in floating-point performance. Our proposed Resonant Shrinkage loss and Output Adaptation can effectively reduce the quantization sensitivity of activations, especially across different task branches within the detection head. Additionally, the proposed Edge-Aware KURE loss further adjusts the weight distribution with fewer outliers. Extensive experiments demonstrate that OAR significantly improves the performance of quantized detectors, particularly under low-bit conditions. This work opens up new possibilities for quantization regularization on weights and activations.

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
