# OpenReview forum: "OAR: Training Quantization-Friendly Object Detectors via Outlier-Aware Restriction"
_ICLR.cc/2026/Conference — Submitted to ICLR 2026_

### Official Review · Reviewer_7au4 · 2025-10-16

**Soundness:** 2
**Presentation:** 1
**Contribution:** 2
**Rating:** 2
**Confidence:** 3

**Summary:**

This paper proposes OAR (Outlier-Aware Restriction), a training framework to improve quantization-friendliness of object detectors. The method introduces three components: Resonant Shrinkage loss to constrain activation boundaries, Output Adaptation with learnable scaling factors for task branches, and Edge-Aware KURE loss for weight regularization. The method is evaluated on both Post-Training Quantization (PTQ) and Quantization-Aware Training (QAT) pipelines, showing improvements in various low-bit scenarios.

**Strengths:**

- Identifies an interesting problem of activation distribution discrepancies in detection heads

- Provides comprehensive ablation studies showing component contributions

**Weaknesses:**

- The core premise is flawed. Modifying pre-training with quantization-oriented losses is a form of model re-training. This contradicts the standard, widely-accepted definition of PTQ, which performs calibration post-training without updating model weights. The work fails to justify why this pre-training approach is preferable to directly using the proposed losses in a QAT setting.

-  The paper's evaluation relies on a weakly-defined "naive PTQ" baseline. It fails to compare against established PTQ calibration methods such as OMSE, Percentile or entropy-based observers.

- The paper lacks comparisons with state-of-the-art PTQ methods (e.g., BRECQ, AdaRound) and modern detector architectures (e.g., Deformable DETR, DINO).

**Questions:**

See Weaknesses.

---

### Official Review · Reviewer_5sTY · 2025-10-30

**Soundness:** 2
**Presentation:** 2
**Contribution:** 2
**Rating:** 2
**Confidence:** 4

**Summary:**

This paper introduces OAR (Outlier-Aware Restriction), a training framework designed to improve the robustness of object detectors under low-bit quantization. It addresses activation and weight outliers by proposing Resonant Shrinkage Loss, Output Adaptation, and Edge-Aware KURE Loss, leading to more quantization-friendly models. Experiments on the MS COCO dataset demonstrate significant performance improvements, especially at 4-bit quantization, with minimal calibration data, making it effective for real-world deployment.

**Strengths:**

1. The authors highlight an interesting insight: the statistical distribution differences between the two, which could significantly impact the quantization process. This observation adds valuable depth to the understanding of model performance under quantization.

2. The method presented in the paper is clearly written and easy to understand, with well-defined formulas that effectively serve the motivation. The clarity of the approach helps convey the core ideas in a straightforward manner, making it accessible while maintaining technical rigor.

**Weaknesses:**

1. In Section 3.3, where the question mark appears incorrectly. It seems to be a citation mistake.
2. Wouldn't it be beneficial to include methods like SmoothQuant and QuaRot in the baseline comparison? These PTQ techniques are effective in handling outliers in large models and have shown good results in mitigating outliers during post-training quantization. Since the proposed approach is training-based and involves a longer time cost, should it not ideally achieve better or more stable performance compared to these PTQ methods? Or could OAR further enhance the performance of these methods?
3. The paper does not provide information on the training time and convergence speed of OAR. Would introducing OAR impact the model’s training convergence? Specifically, does the training time increase significantly, and does OAR affect the speed at which the model converges compared to traditional methods?
4. The experimental results show only marginal improvements in quantization performance, with some settings (e.g., QDrop with RetinaNet-ResNet50-INT2) performing better. There are also cases where the performance is worse than the FP32 baseline. What could explain these inconsistencies?

**Questions:**

See weaknesses

---

### Official Review · Reviewer_NkUe · 2025-11-05

**Soundness:** 2
**Presentation:** 2
**Contribution:** 2
**Rating:** 2
**Confidence:** 4

**Summary:**

This paper introduces Outlier-Aware Restriction (OAR), a training framework aiming to improve quantization robustness for object detectors by regularizing activations and weights. The method comprises three main components: (1) Resonant Shrinkage loss for activation stabilization, (2) Output Adaptation for branch-wise scaling, and (3) Edge-Aware KURE loss for weight regularization. Experiments on MS COCO show improvements under low-bit quantization (especially 4-bit and 2-bit) for several detectors (RetinaNet, ATSS, YOLOX-tiny).

**Strengths:**

1. The framework is conceptually simple and compatible with existing quantization methods.
2. The experime ntal section is fairly detailed, covering multiple detectors and quantization schemes.

**Weaknesses:**

1. Overall, the work combines existing ideas rather than presenting a clear new theoretical insight or algorithmic innovation.
2. Weak theoretical grounding. The methodology lacks a solid theoretical justification for why these specific regularizers should directly lead to better quantization robustness. The intuition about “outliers” is plausible but never formalized or empirically analyzed beyond heuristic evidence.
3. Experimental analysis lacks rigor. Improvements in mAP are relatively small and inconsistent across models (e.g., many results fluctuate within 1–2 mAP). Comparisons are limited to a few older baselines (e.g., LSQ, QDrop); more recent and stronger quantization approaches are missing.

**Questions:**

See the weaknesses.

---

### Meta-Review · Area_Chair_DJZs · 2026-01-07

**Summary:**

NkUe	Weak theoretical grounding and experimental analysis lacks rigor. Comparison with recent methods is lacking
5sTY 	Experimental results show only marginal improvements and sometime worse than baseline. Reviewer has asked for explanation on these inconsistencies and questioned if other comparisons for baseline will be beneficial.
7au4	Questions the premise of the work. Not satisfied with how the baseline is setup.  Points out that the comparisons with many SOTA methods are not there.

**Reviewer Concerns:**

None.

**Reviewer Scores:**

Unless rebuttal has been quiet thorough, discussion might not have helped.

---

### Decision · Program_Chairs · 2026-01-26

Reject